# Copiotrophy in a Marine-Biofilm-Derived Roseobacteraceae Bacterium Can Be Supported by Amino Acid Metabolism and Thiosulfate Oxidation

**DOI:** 10.3390/ijms24108617

**Published:** 2023-05-11

**Authors:** Xiaoyan Su, Han Cui, Weipeng Zhang

**Affiliations:** College of Marine Life Sciences, Ocean University of China, Qingdao 266003, China; suxiaoyan@stu.ouc.edu.cn (X.S.); cuihan@stu.ouc.edu.cn (H.C.)

**Keywords:** copiotrophy, Roseobacteraceae, biofilm

## Abstract

Copiotrophic bacteria that respond rapidly to nutrient availability, particularly high concentrations of carbon sources, play indispensable roles in marine carbon cycling. However, the molecular and metabolic mechanisms governing their response to carbon concentration gradients are not well understood. Here, we focused on a new member of the family Roseobacteraceae isolated from coastal marine biofilms and explored the growth strategy at different carbon concentrations. When cultured in a carbon-rich medium, the bacterium grew to significantly higher cell densities than *Ruegeria pomeroyi* DSS-3, although there was no difference when cultured in media with reduced carbon. Genomic analysis showed that the bacterium utilized various pathways involved in biofilm formation, amino acid metabolism, and energy production via the oxidation of inorganic sulfur compounds. Transcriptomic analysis indicated that 28.4% of genes were regulated by carbon concentration, with increased carbon concentration inducing the expression of key enzymes in the EMP, ED, PP, and TCA cycles, genes responsible for the transformation of amino acids into TCA intermediates, as well as the *sox* genes for thiosulfate oxidation. Metabolomics showed that amino acid metabolism was enhanced and preferred in the presence of a high carbon concentration. Mutation of the *sox* genes decreased cell proton motive force when grown with amino acids and thiosulfate. In conclusion, we propose that copiotrophy in this Roseobacteraceae bacterium can be supported by amino acid metabolism and thiosulfate oxidation.

## 1. Introduction

Carbon is essential for the survival of microorganisms. In marine environments, the carbon concentration plays a major role in determining both the growth and niche distribution of bacteria. This is especially true for chemoheterotrophic bacteria that obtain carbon entirely from organic compounds. Marine heterotrophic bacteria can be divided into oligotrophs and copiotrophs according to their nutritional strategy [1]. Oligotrophic bacteria grow slowly in all types of carbon sources, with high concentrations of organic materials often adversely affecting their growth, whereas copiotrophic bacteria respond rapidly to nutrient availability and can grow well in media with high concentrations of organic matter [2]. Moreover, copiotrophic bacteria are genetically more capable of sensing, transducing, and integrating extracellular stimuli [1,2]. These characteristics may be key to their ability to fine-tune and respond rapidly to changes in environmental conditions, such as a sudden influx or depletion of nutrients [1], and allow for them to degrade and utilize high-molecular weight particulate organic materials in the marine ecosystem [3]. However, the molecular and metabolic mechanisms governing the response of copiotrophic bacteria to changes in carbon source concentrations are not well understood.

Bacterial members of the family Roseobacteraceae, previously known as the *Roseobacter* group and belonging to the family Rhodobacteraceae [4], are a phylogenetically uniformed and physiologically heterogeneous group within the class Alphaproteobacteria [5,6,7]. They are widely distributed in the ocean, ranging from offshore waters to the open ocean, from surface waters to the deep sea, and from the tropics to the polar regions [6,7,8], indicating that Roseobacteraceae have adapted to a variety of carbon sources and concentrations. Thus, an understanding of carbon metabolism and trophic strategies in Roseobacteraceae bacteria is of particular significance. *Ruegeria pomeroyi* DSS-3, previously known as *Silicibacter pomeroyi* DSS-3 [9], is the first Roseobacteraceae bacterium with a complete genome sequence [10] and is used as a model microorganism for the study of the eco-physiological strategies of heterotrophic and free-living bacteria [11]. Moreover, DSS-3 has been defined as a moderate copiotroph based on genomic and preliminary physiological evidence [12]. 

In addition to free-living lifestyles, several Roseobacteraceae lineages prefer living in biofilms [13]. In coastal marine environments, biofilms can develop on a variety of substrates, including natural stones, sedimental particles, and various man-made materials [14,15], and they often possess great microbial diversity [16,17]. In a recent study [18], we systematically explored the diversity and thiosulfate metabolism features of Roseobacteraceae strains in marine biofilms and highlighted their important roles in biogeochemical cycles. On the other hand, coastal marine biofilms are believed to be dominated by copiotrophic bacteria, attributed to the relatively higher concentrations of organic carbon in these waters compared with the open ocean [19,20]. However, due to the complexity and high species diversity of marine biofilms, biofilm-associated bacteria that perform copiotrophic lifestyles have not been well-studied.

In the present study, we explored the copiotrophic strategy of marine-biofilm-associated Roseobacteraceae bacterium, focusing on one fast-growing strain isolated from biofilms on coastal stone surfaces. The growth and biofilm formation of this bacterium were examined in the presence of different carbon source concentrations, using *Ruegeria pomeroyi* DSS-3 as a reference strain. We then adopted genomic, transcriptomic, and metabolomic analyses, as well as physiological and biochemical experiments, to identify specific genes and pathways that contribute to the response of this bacterium to variations in carbon source concentrations. 

## 2. Results

### 2.1. Overall Genomic Information and Phylogeny

The strain was isolated from a rock surface biofilm community immersed in coastal water at Qingdao, China, during the conduction of our microbial biofilm genomics project. After preliminary identification based on the Sanger sequencing of the 16S rRNA gene, its complete genome was sequenced using the PacBio sequencing technique. The overall genomic information, including genome size, GC content, the number of plasmids, number of rRNA and tRNA genes, and number of open reading frames (ORFs), is shown in Appendix A. The bacterium possessed one chromosome and three plasmids, composed of 4760 ORFs in total. Genome searching against the GTDB-TK database [21] assigned the bacterium to an unclassified genus, and searching against the NCBI database using its 16S rRNA gene assigned this bacterium as a member of Roseobacteraceae. Thus, the bacterium was described as *Roseobacteraceae* sp. M382 (the 382nd strain isolated from marine biofilm samples). A phylogenetic analysis including house-keeping genes derived from M382 and the available Roseobacteraceae genomes supported the relatively close relationship between M382 and *Ruegeria* strains, including *R. pomeroyi* DSS-3; however, it was apparent that M382 formed an independent branch in the tree (Figure 1). Whole-genome average nucleotide identity (ANI) analysis revealed that M382 and DSS-3 shared 74.46% identity, which was lower than the identities between DSS-3 and other *Ruegeria* strains (*Ruegeria* sp. AD91A and *Ruegeria* sp. THAF33). Thus, it is likely that M382 belongs to a previous-unstudied genus, which, however, needs to be further confirmed in future taxonomic studies. In addition, it is noteworthy that M382 has a larger genome size (5.20 Mb) than its close relatives, including DSS-3 (4.60 Mb), AD91A (4.41 Mb), THAF33 (4.63 Mb), and *Leisingera aquaemixtae* R2C4 (4.38 Mb) (Appendix A).

### 2.2. Copiotrophic Growth Features

As mentioned above, M382 was found to have a larger genome size than DSS-3, which is a copiotrophic bacterium [12], thus motivating us to hypothesize that M382 may also display copiotrophic features. To test this hypothesis, we cultured M382 and DSS-3 in marine broth 2216E media with different concentrations of carbon (original carbon concentration versus 1/10 carbon concentration). After being cultured at 25 °C for 24 h, M382 growth reached the stationary stage with OD_600_ values of approximately 1.4 and 0.4 for the original (labeled as 2216E) and 1/10 carbon concentrations (1/10 2216E), respectively, while the growth of DSS-3 was not profoundly affected by the carbon concentration (Figure 2A,B). Given that M382 was isolated from a biofilm community, we also investigated its growth as a biofilm. As expected, M382 formed thick biofilms on the bottom of 6-well plates when cultured without shaking. In contrast, no biofilm was observed for DSS-3, consistent with a previous study that defined it as exclusively free-living [22]. The biofilm structures of M382 under different carbon concentrations were observed using a scanning electron microscope (SEM) (Figure 2C,D) and the cell densities were quantified using a microplate reader (Figure 2E). All the results indicated copiotrophic growth characteristics.

### 2.3. Genomic Pathway Reconstruction 

With the genome in hand, we next performed KEGG annotation and pathway reconstruction to evaluate the metabolic potential of M382. The major pathways for carbon and energy metabolism are shown in Figure 3. The Embden–Meyerhof–Parnas (EMP) pathway, the Entner–Doudoroff (ED) pathway, and the pentose phosphate (PP) pathway were identified (Figure 3). These pathways convert glucose to acetyl-CoA, which then enters the tricarboxylic acid (TCA) cycle for the generation of energy (NADH, ATP, and GTP) and intermediate products. The generated NADH may be directed to the cell membrane for ATP production through I-V respiratory chain complexes (Figure 3). Acetyl-CoA was also found to be linked to fatty acid metabolism through a number of genes in M382 (Figure 3). Notably, M382 possessed a variety of genes and pathways responsible for amino acid metabolism, including enzymes for deamination (e.g., D-serine dehydratase *dsdA*) and transamination (e.g., aspartate aminotransferase) (Figure 3). The respective enzymes catalyze the transformation of amino acids to various intermediate products (e.g., oxaloacetate) that link amino acid and central carbon metabolic pathways. In addition, transporters involved in the general uptake of amino acids (i.e., *aap* gene family) and the uptake of branch-chained amino acids (i.e., *liv* gene family) were identified (Figure 3). In terms of nitrogen metabolism, a complete denitrification pathway was found (Figure 3), from the reduction of nitrate to nitrite by *napAB* to the reduction of nitrous oxide to nitrogen gas by *nosZ*. Additionally, M382 possessed several copies of glutamine synthetase (*glnA*, K01915) and glutamate dehydrogenase (*gdhA*, K00261), which converts ammonia to glutamine and glutamate, respectively (Figure 3). In terms of sulfur metabolism, the *sox* pathway that oxidizes thiosulfate to sulfate was identified (Figure 3), which can be used for energy production in addition to the oxidation of organic carbon compounds. The *sat* and *cys* genes involved in the assimilatory reduction of sulfate to sulfide were also identified (Figure 3). The identification of these pathways in the M382 genome suggested that (1) the bacterium was a heterotrophic microbe that utilizes a number of amino acids as carbon sources; (2) carbon metabolism in the bacterium was linked with nitrogen or sulfur metabolism for energy production.

The M382 genome was then compared with four strains that were shown by phylogenetic analysis to be closely related, namely, *R. pomeroyi* DSS-3, *Ruegeria* sp. THAF33, *Rue*g*eria* sp. AD91A, *Leisingera aquaemixtae* R2C4, and *Rue*g*eria* sp. AD91A. KEGG analysis showed that M382 had 116 unique genes (Figure 4A and see Appendix A for the full list of these unique genes). According to the KEGG database, these unique genes could be summarized into 58 categories, including biofilm formation, transcription regulation, and amino acid metabolism (Figure 4B). For example, genes associated with the general secretory pathway (e.g., *gspD*, K02453; *gspH*, K02456; *gspK*, and K02460) have been demonstrated to be involved in bacterial biofilm formation [23], hydroxycarboxylate dehydrogenase B (*hcxB*, K13574) is involved in tyrosine metabolism [24], and homoserine O-acetyltransferase (*metX*, K00641) is involved in cysteine and methionine metabolism [25], while dimethylglycine N-methyltransferase (*sdmt*, K18897) functions in glycine, serine, and threonine metabolism [26], and 5-methylthioadenosine (*mtaD*, K12960) functions in cysteine and methionine metabolism [27]. These unique genes are likely to support the adaptation of M382 to the biofilm lifestyle together with the capacity to metabolize amino acids.

### 2.4. Gene Expression in Response to Carbon Source Concentration

To elucidate the molecular mechanisms supporting copiotrophy, we performed transcriptomic analyses of M382 biofilms grown in 2216E and 1/10 2216E, with three replicates for each condition. The transcriptomic reads were mapped to the ORFs of M382 and the reads per kilobase of exon per million reads mapped (RPKM) values were calculated. In total, 13.98 Gb of transcriptomic data were obtained; detailed information, including read numbers, read lengths, and read quality, is provided in Appendix A. Based on statistical analyses of the RPKM values, the expression of 1352 genes (28.4% of the total number of genes) differed significantly (*p*-value < 0.05 in the two-tailed Student’s *t*-test and RPKM fold-change >2) between the two culture conditions. Of these 1352 genes, 994 were annotated by the KEGG database (Appendix A). Most of these KEGG-annotated genes were associated with carbon metabolism and energy production. The 54 genes associated with central carbon metabolism, including the EMP, ED, PP, and TCA cycles, and amino acid transport and transformation, are shown in Figure 5. Nearly all the differentially expressed genes associated with the EMP pathway were down-regulated by a high carbon concentration, such as glyceraldehyde 3-phosphate dehydrogenase (GAPDH, K00134) which catalyzes the sixth step of EMP, and three key genes of the PP pathway, namely, glucose-6-phosphate 1-dehydrogenase (*zwf*, K00036), 2-dehydro-3-deoxyphosphogluconate aldolase (*eda*, K01625), and phosphogluconate dehydratase (*edd*, K01690) (Figure 5). In the TCA cycle, the enzyme involved in the first step (pyruvate dehydrogenase, K00163, which catalyzes pyruvate to form acetyl-CoA) was down-regulated while all the differentially expressed genes were up-regulated by a high carbon concentration (Figure 5). Notably, various genes involved in amino acid metabolism, such as glycine cleavage system A protein (K02437), cysteine synthase (K01738), glutamate N-acetyltransferase (K00620), and arginase (K01476), were up-regulated by a high carbon concentration (Figure 5). In addition, a number of differentially expressed genes were associated with sulfur metabolism, exemplified by *soxABCXYZ*, and were up-regulated by a high carbon concentration (Figure 5).

### 2.5. Cellular Metabolic Profiling and Preferential Utilization of Amino Acids 

As both the genomic and transcriptomic results suggested that amino acid metabolism and its associated pathways were regulated by carbon concentrations in M382, we hypothesized that there should be higher amino acid concentrations in bacterial cells cultured with higher carbon concentrations due to the preferential utilization of amino acids and the higher metabolic rate. To test this hypothesis, we cultured the M382 biofilms in 2216E and 1/10 2216E media and measured the intracellular amino acid concentrations (µg/g) using metabolomics. Significantly higher levels (*p*-value < 0.05) of L-histidine, L-serine, L-arginine, glycine, L-aspartic acid, L-threonine, L-glutamic acid, L-alanine, L-proline, L-methionine, L-valine, L-ornithine, L-lysine, L-isoleucine, L-leucine, L-phenylalanine, and L-tryptophan were found in M382 cells cultured in the high-carbon 2216E medium in comparison with those cultured in the 1/10 carbon 2216E medium (Figure 6). More specifically, in M382 cultured in 2216E, glutamate, aspartate, and alanine were the top three amino acids ranked by concentration (Figure 6).

Considering that bacterial amino acid metabolism can be up-regulated in complex media [28], we investigated M382 growth in the presence of amino acids or saccharides to test the specificity of amino acid consumption. M382 biofilms were cultured in media with two concentrations of mixed amino acids (L-glutamic acid, glycine, L-tryptophan, L-alanine, L-threonine, L-cysteine, and L-serine) and the maximum cell densities were measured. In parallel, M382 biofilms were cultured with two concentrations of saccharides, which were used for comparison. As expected, after 24 h of culture, the M382 biofilm grew to a cell density of OD_600_ = 0.74 in the 5 mM amino acid mixture and a cell density of OD_600_ = 0.24 in the 0.5 mM amino acid mixture (Figure 7). The cell density of the M382 biofilm grown in 5 mM saccharides was OD_600_ = 0.167, compared to OD_600_ = 0.096 in the 0.5 mM saccharide-containing medium. Although significant differences in cell densities were seen with both amino acids and saccharides, the fold-change along with concentration variation was much greater for bacteria grown in amino acids (Figure 7).

### 2.6. Association between Amino Acid Metabolism and Sulfur Oxidation

The above results demonstrated an association between amino acid concentration and the growth and metabolism of M382. We wondered whether amino acid utilization is related to other forms of energy production, such as the oxidation of inorganic compounds. Note that the transcription of the *sox* gene cluster was induced by higher carbon concentration in M382 (Figure 5), and it is documented that this gene cluster is involved in thiosulfate oxidation and energy production. Thus, we hypothesized that thiosulfate oxidation was related to amino acid metabolism in M382. To test this hypothesis and further elucidate the response of M382 to high carbon concentrations, we obtained two mutants of the *sox* genes (*soxA* and *soxX*). The wild-type strain and the two mutants were grown as biofilms in amino acid-containing media (5 or 0.5 mM of amino acid mixture) with thiosulfate, and the proton motive force (PMF) was measured after 24 h. In media containing a 5 mM amino acid mixture as the carbon source, the wild-type M382 showed significantly higher PMF values than Δ*soxA* (two-tailed Students’ *t*-test, *p*-value < 0.05) or Δ*soxX* (two-tailed Students’ *t*-test, *p*-value < 0.01) (Figure 8). In contrast, strains in the 0.5 mM of amino acid mixture displayed no significant differences in terms of PMF. These results suggested that this bacterium rely on thiosulfate oxidation for energy production only in the presence of higher concentration of amino acids, and thus thiosulfate oxidation is likely to be correlated with the utilization of high concentrations of amino acids, resulting in an efficient energy production and fast growth.

## 3. Discussion

In the present study, we discovered the copiotrophic characteristics of a Roseobacteraceae bacterium derived from coastal marine biofilms. By comparing *Roseobacteraceae* sp. M382 with *R. pomeroyi* DSS-3, we identified it as a copiotroph. M382 probably represents a new genus that is phylogenetically close to *Ruegeria*, indicating that physiological comparisons between M382 and DSS-3, a known copiotroph, are reasonable. It is apparent that the carbon concentration in the media has a profound effect on the growth of M382. M382 grew faster and accumulated more biomass than DSS-3 when grown in the presence of a high carbon concentration. Moreover, complete genome sequencing indicated that M382 has a larger genome than DSS-3, and it is known that copiotrophs often have large genomes as they possess more functional genes for transporting or metabolizing carbon sources, reflecting a high degree of flexibility in their response to environmental circumstances [29,30]. In addition, a comparison between M382 and its four close relatives revealed unique genes belonging to transcription factors, also in congruent with the notion that diverse regulatory genes are often detected in the genomes of copiotrophs [29,30].

Multi-omics analyses, including genomics, transcriptomics, and metabolomics, demonstrated that the response of M382 to carbon concentration is largely dependent on amino acid metabolism. Genomic comparisons revealed the presence of many unique genes that were related to the metabolism of various amino acids as well as genes that contributed to biofilm formation. The presence of these unique genes may be the result of the biofilm lifestyle, as marine biofilms have been suggested to be a hotspot for the bacterial acquisition of useful genes [31,32]. These results also implied that amino acid metabolism is adopted by M382. Transcriptomics indicated significant alterations in the expression of 28.4% of genes in response to the environmental carbon concentration, and a number of these genes were responsible for amino acid metabolism. These results highlighted the importance of amino acids in supporting copiotrophy in M382. Moreover, the gene transcription profiles at the high carbon concentration suggested the likelihood of the TCA cycle being directly driven by amino acids, rather than by pyruvate from the EMP, ED, or PP routes. This notion is based on the up-regulation of metabolic pathways related to glutamate, aspartate, and alanine, which can be transformed into 2-oxoglutaric acid, oxaloacetate, and pyruvate, respectively, and enter the TCA cycle. In contrast, genes from the EMP, ED, and PP pathways were observed to be down-regulated at a high carbon concentration. Consistently, the metabolomics results showed that M382 accumulates higher amino acid contents when grown in high concentrations of mixed carbon sources, suggesting the selective consumption of amino acids from the mixed carbon sources. Notably, glutamate, aspartate, and alanine were the top three amino acids ranked by concentration, in line with the transcriptomics results showing the up-regulation of genes related to the metabolism of these three amino acids. In addition, a stronger influence of amino acid concentration on bacterial growth than the saccharide concentration was observed, which is also consistent with the notion that the copiotrophic growth of M382 can be supported by amino acid metabolism.

The demonstrated correlation between thiosulfate oxidation, amino acid metabolism, and energy production further supports the role of amino acids in supporting copiotrophy in M382. It is well known that in autotrophic bacteria, especially bacteria living in deep-sea environments, thiosulfate oxidation provides a large proportion of the energy requirements [33]. Although direct evidence is lacking, it is possible that thiosulfate oxidation is also associated with energy production in heterotrophic bacteria. We addressed this question by measuring the PMF of the wild-type M382 strain as well as two *sox* mutants. PMF is an electrochemical proton gradient across the cytoplasmic membrane that drives vital processes in cells such as ATP synthesis and the transport of a wide range of substrates [34]. The higher PMF values observed for the wild-type strain in higher amino acid concentrations point to an association between the metabolism of these carbon and sulfur elements. Moreover, these findings are consistent with the transcriptomics results showing up-regulation of the *sox* gene cluster by high carbon source concentration.

## 4. Materials and Methods

### 4.1. Sampling and Bacterial Strain Isolation

The biofilm samples used for bacterial isolation were collected from immersed subtidal rocky surfaces in the coastal seawaters of Qingdao (120.145, 39.915). Loosely attached bacterial cells were washed off before scraping off the biofilm with sterilized cotton swabs. The scraped biofilm was vortexed and suspended in sterile seawater before serial dilution (10, 10^2^, 10^3^, 10^4^, and 10^5^-fold dilutions) and plating on 2216E marine agar plates. The plates were incubated in an incubator at 25 °C. Single colonies growing on the agar plates were picked after five generations, followed by 16S rRNA gene amplification in a Prime Star Max premix (Takara, Beijing, China) with the primer pair 27F: 5′-AGAGTTTGATCCTGGCTCAG-3′ and 1492R: 5′-GGTTACCTTGTTACGACTTC-3′. Sanger sequencing of the PCR products was performed by Sangon Biotech (Shanghai, China), and the sequences were used as queries to search against the NCBI Nucleotide database for taxonomic identification.

### 4.2. Genome Sequencing and Analyses

The whole genome of M382 was sequenced on the PacBio CCS and the Illumina NovaSeq platforms by Novogene (Tianjin, China). The complete genome was obtained by mix-assembly of the PacBio and Illumina reads using SPAdes 3.0.0 [35] installed on a local Linux system. Prodigal 2.6.3 [36] installed on the local Linux system was used to predict the ORFs and corresponding protein sequences. To generate the phylogenetic tree, 28 housekeeping genes (*frr*, *infC*, *nusA*, *pgk*, *pyrG*, *rplA*, *rplB*, *rplC*, *rplD*, *rplE*, *rplF*, *rplK*, *rplL*, *rplM*, *rplN*, *rplP*, *rplS*, *rplT*, *rpmA*, *rpsB*, *rpsC*, *rpsE*, *rpsI*, *rpsJ*, *rpsK*, *rpsM*, *rpsS*, *tsf*) were extracted from M382 and other 94 Roseobacteraceae strains, whose complete genomes are available in NCBI. The maximum-likelihood phylogenetic tree of the linked housekeeping genes was constructed using MEGA 7.0 [37] using a neighbor-joining model with 1000 bootstrap replicates. ANI between M382 and DSS-3 was analyzed using the online ANI Calculator of EZBioCloud [38]. Functional annotation of the genes was performed using M382 protein sequences in BLASTp (e-value set to 1 × 10^−7^) and in the KEGG database (2022 updated version) [39]. For pathway analysis, the online software KEGG Mapper (https://www.genome.jp/kegg/mapper.html)(accessed on 1 June 2022) was used to reconstruct the functional pathways based on the annotated proteins to assess potential metabolic capacities.

### 4.3. Bacterial Growth and Biofilm Formation

To evaluate the influence of carbon concentration on the growth rate and maximum cell densities of DSS-3 and M382, the bacteria were cultured in marine broth 2216E (Difco, Franklin Lakes, NJ, USA) and 1/10 2216E for 44 h at 25 °C in an automated microbiological growth curve analysis system (Bioscreen C, Labsystem, Paris, France). To avoid the influence of minerals and salinity, the inorganic nutrients in 1/10 2216E were adjusted to the same concentrations as those in 2216E. 

For biofilm formation and cell density quantification in 2216E media, M382 cultures with OD_600_ = 1.0 were incubated in 2216E and 1/10 2216E in 6-well plates. After incubation at 25 °C for 24 h, almost all the cells were attached to the bottom of the plates to form thick biofilms. The cells were then scattered by ‘blowing and sucking’ using a 1 mL pipette, and cell densities were measured using a Biotek Cytation5 imaging reader (Biotek Instruments, Winooski, VT, USA) at a wavelength of 600 nm. For phenotype observation, biofilms were washed twice with PBS and fixed with 2.5% glutaraldehyde before dehydration in isoamyl acetate and vacuum plating. The morphology of cells was observed by using a SEM system (TESCAN VEGA3, Brno, Czechoslovakia) at Qingdao University. For biofilm formation and cell density quantification in synthetic media with amino acids, equal concentrations of L-glutamic acid, glycine, L-tryptophan, L-alanine, L-threonine, L-cysteine, and L-serine, each at a concentration of 5 mM or 0.5 mM, were mixed and used as carbon sources. For the experiment using synthetic media with saccharides, equal concentrations of sucrose, L-glucose, D-glucose, L-fructose, D-fructose, D-maltose, and D-galactose, each at a concentration of 5 mM or 0.5 mM, were used as carbon sources. Again, inorganic nutrients were adjusted to the same concentrations. The same culture conditions and methods as described above were used for biofilm formation and cell density quantification.

### 4.4. Transcriptomic Sequencing and Analyses 

M382 biofilms were cultured in 2216E and 1/10 2216E media in 6-well plates at 25 °C for 24 h. The biofilms formed at the bottom of the plates were scattered using a 1 mL pipette and transferred to 50 mL Falcon tubes. The bacterial cells in the Falcon tubes were then centrifuged at 4500 rpm and the precipitates were immediately transferred to liquid nitrogen. RNA extraction was conducted using the RNAprep Pure Cell/Bacteria Kit (Tiangen Biotech, Beijing, China) and libraries were prepared using the NEBNext Ultra RNA Library Prep Kit (NEB, Ipswich, MA, USA). Transcriptomic sequencing was then performed on the Novaseq platform by Novogene Bioinformatics (Beijing, China). Raw transcriptomic data were quality-controlled using the NGS QC Toolkit v2.3.3 [40] with default parameters. After building an index based on all ORFs, Bowtie v2.4.2 [41] and Samtools v1.11 [42] were used to calculate and view gene expression levels by calculating the RPKMs. The significance of differences in gene expression was evaluated using Student’s *t*-test, with *p*-value < 0.05 plus RPKM fold-change >2 as the cutoff. 

### 4.5. Quantitative Determination of Cellular Amino Acids

M382 biofilms were cultured in 2216E and 1/10 2216E media in 6-well plates at 25 °C for 24 h. The cells were harvested and stored using the same procedures as those used in the preparation for transcriptomic sequencing. UHPLC-MS/MS analysis was performed on an Agilent 1290 Infinity II UHPLC system linked to a 6470A Triple Quadrupole mass spectrometry (Santa Clara, CA, USA) at the Profleader Biotech Company (Shanghai, China). The weighted samples were spiked with 10 mL of 6M HCl and 0.5% β-mercaptoethanol, sealed under nitrogen, and hydrolyzed at 110 °C for 22 h. After hydrolysis, 100 μL of the hydrolysate was dried under nitrogen and HCl traces were removed by deionized water. The samples were then dried and dissolved in 50 μL of sodium carbonate (100 mM) and derivatized by the addition of 50 μL of 2% benzoyl chloride. The derivatized samples were isometrically mixed with internal standard solutions before UHPLC-MS/MS analysis. The raw data were processed by MassHunter Workstation Software (version B.08.00, Agilent, Santa Clara, CA, USA) with default parameters. Calibration curves of ten points were constructed by plotting the peak area ratio of each compound to the internal standard against the concentration of each compound. 

### 4.6. Gene Knockout

Gene knockout was performed based on in-frame mutant construction and complementation [43,44], with modification. The pHG1.0 plasmid hosted by *Escherichia coli DB3.1* was used for in-frame deletion of *soxA* or *soxX*. First of all, PCR primers (*soxA* Up-F, *soxA* Up-R, *soxA* Down-F, *soxA* Down-R, *soxX* Up-F, *soxX* Up-R, *soxX* Down-F, and *soxX* Down-R; Appendix A) were designed to amplify two regions flanking the upstream and downstream regions of the target genes. The two flanking regions of a gene were fused through a complementary linker region. The size of the flanking regions serving for homologous recombination is 29 bp and these regions are marked red in Appendix A. The sequences of Up-F and Down-R also contained regions for ligation with the pHG1.0 plasmid. The ‘mutated copy’ of a target gene was generated by PCR using the primers Up-F and Down-R. The PCR product was inserted into the empty pHG1.0 plasmid using Gateway BP clonase II enzyme mix (Thermo Fisher, Waltham, MA, USA). After incubation for two hours at 25 °C, the ligated DNA product was transformed into *E. coli* wm3064 competent cells through heat shocking at 42 °C for 30 s. The strains were then plated on LB agar with 50 µg/mL of gentamicin and 50 µg/mL of 2,6-Diaminopimelic acid. After cultivation for 16 h at 37 °C, successfully transformed cells were detected using PCR. DNA conjugation between *E. coli* wm3064 carrying the ‘mutated copy’ DNA fragment and M382 were then conducted following three steps. First, these two strains were grown in LB medium with 50 µg/mL of gentamicin and marine broth 2216E with 50 µg/mL of 2,6-Diaminopimelic acid, respectively. Second, 2 mL of wm3064 and 1 mL of M382 were mixed, harvested by centrifugation at 4000 rpm, resuspended in 200 µL of marine broth 2216E medium, and plated on marine broth 2216E medium with 50 µg/mL of gentamicin. Third, 500 colonies were screened by PCR for successful recombination. Finally, candidate bacterial cells were transferred to marine broth 2216E medium with 20% sucrose, where colonies with two-step recombination were generated and screened by PCR using the Up-F and Down-R primers. This step is to screen colonies that have successfully resected the suicide plasmid, which contains the *sacB* gene and cannot grow in sucrose. The number of successful clones obtained were approximately half the number of clones obtained in the first recombination.

### 4.7. Measurement of Membrane Potential

Biofilms of the wild-type M382 and the two *sox* mutants (Δ*soxA* and Δ*soxX*) were grown in synthetic media with or without 10 mM of thiosulfate. They were cultured for 24 h and harvested by centrifugation at 6000 rpm for 2 min. The bacterial cells were washed twice in PBS (pH 7.4) and resuspended to an OD_600_ of 0.2. The transmembrane potential was measured using a membrane-potential-sensitive probe DiSC3(5), as described in a previous study [45]. DisC3(5) can penetrate the outer membranes of bacteria resulting in a quenching effect. KCl and DiSC3(5) were added to the bacterial solution to final concentrations of 100 mM and 1 μm, respectively. Then, bacterial cells were incubated at room temperature for 25 min in the dark. Fluorescence was detected by using the Biotek Cytation5 imaging reader at an excitation wavelength of 622 ± 10 nm and an emission wavelength of 670 ± 10 nm. 

## 5. Conclusions

Together, the present study has clarified the mechanisms governing copiotrophy in a marine Roseobacteraceae bacterium. We showed that copiotrophy in this bacterium is the result of amino acid metabolism and the regulation of functionally relevant genes. These findings contribute to the understanding of marine bacterial lifestyles and provide an explanation of bacterial adaptation in an ecological framework. Future perspectives would include a further study to illuminate the gene regulatory pathways or networks that control the response to the amino acid concentration, and a broad study to overview similar mechanisms among marine bacteria.

## Figures and Tables

**Figure 1 ijms-24-08617-f001:**
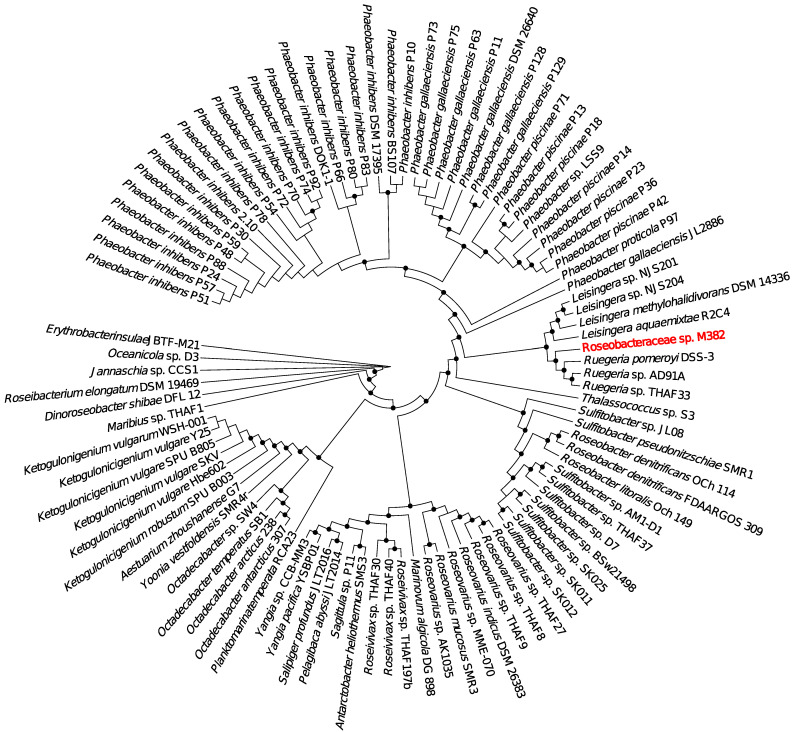
Phylogenetic tree showing the evolutionary relationships between M382 and other representative Roseobacteraceae strains. The tree was built based on the concatenation of 28 housekeeping genes (*frr*, *infC*, *nusA*, *pgk*, *pyrG*, *rplA*, *rplB*, *rplC*, *rplD*, *rplE*, *rplF*, *rplK*, *rplL*, *rplM*, *rplN*, *rplP*, *rplS*, *rplT*, *rpmA*, *rpsB*, *rpsC*, *rpsE*, *rpsI*, *rpsJ*, *rpsK*, *rpsM*, *rpsS*, *tsf*). Bootstrap values were generated based on 1000 replicates. Values above 70% are indicated by black nodes.

**Figure 2 ijms-24-08617-f002:**
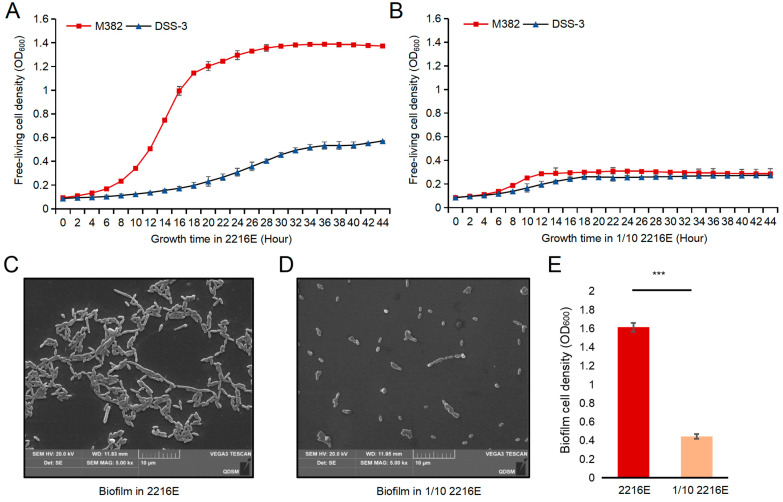
Growth features of M382 and DSS-3 at different concentrations of carbon source. (**A**,**B**) Growth kinetics of M382 and DSS-3 indicated by OD_600_ in 2216E and 1/10 2216E media. (**C**,**D**) Scanning electron micrographs of biofilms formed by M382 in 2216E and 1/10 2216E media. (**E**) Maximum biofilm cell density of M382 grown in 2216E and 1/10 2216E media (*** *p*-value < 0.001 in two-tailed Student’s *t*-test). The error bars represent the standard deviation of three biological replicates.

**Figure 3 ijms-24-08617-f003:**
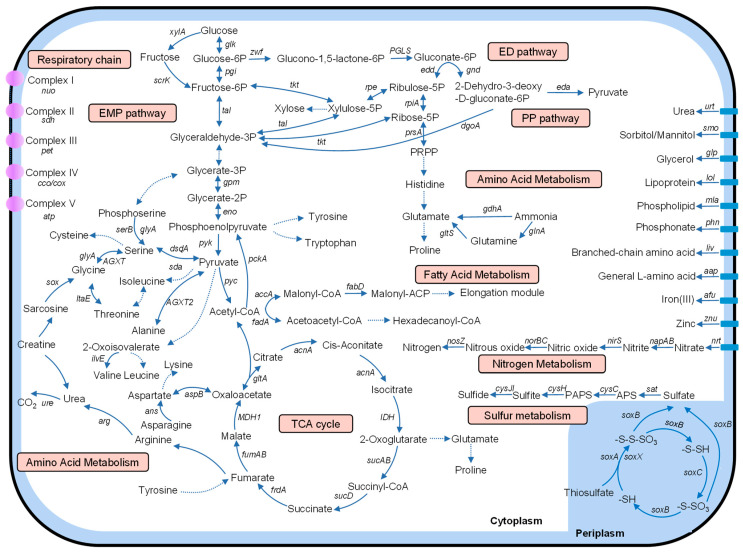
Major pathways of carbon and energy metabolism in M382. Protein sequences were predicted from the complete genome of M382 and annotated using the KEGG database. Metabolic pathway reconstruction based on the KEGG-annotated proteins was performed using KEGG Mapper. The dashed lines represent simplified pathways.

**Figure 4 ijms-24-08617-f004:**
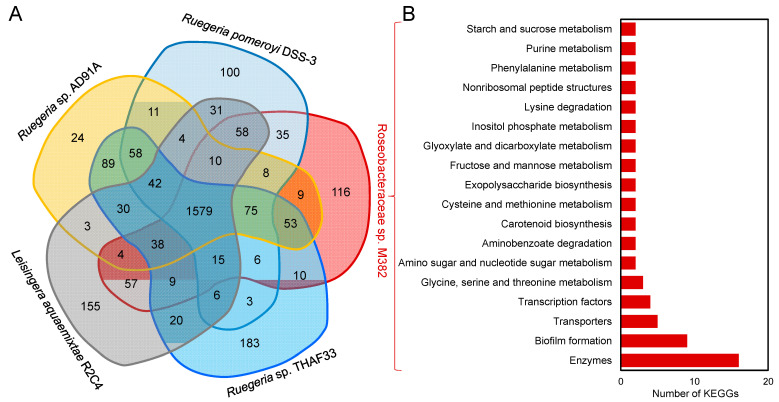
Venn analysis of KEGG-annotated genes between M382 and four closely related strains. These strains were identified in the phylogenetic analysis using house-keeping genes, and all the strains have complete genomes. (**A**) Common and specific KEGG-annotated genes derived from the analyzed genomes. (**B**) Categories of KEGG-annotated genes that are specific to M382. Categories with more than two KEGG-annotated proteins are shown.

**Figure 5 ijms-24-08617-f005:**
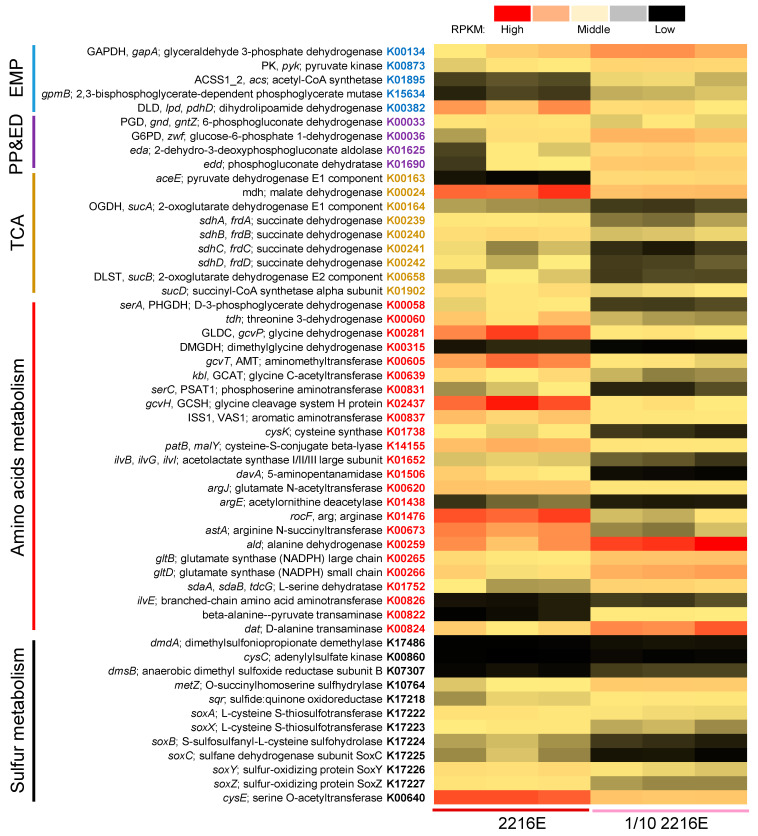
Comparative analysis of M382 biofilms grown at two carbon source concentrations. Significantly altered genes (Student’s *t*-test, *p*-value < 0.05 and RPKM fold-change >2) belonging to central carbon, amino acid, and sulfur metabolism pathways are shown.

**Figure 6 ijms-24-08617-f006:**
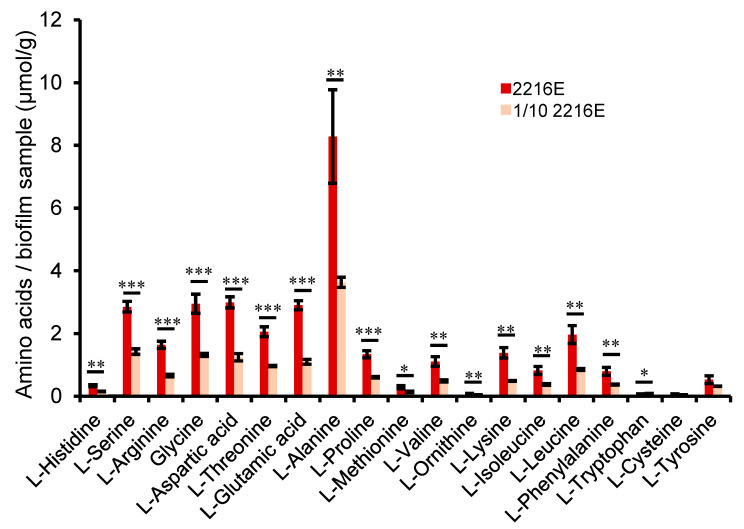
Evidence for amino acid metabolism in M382 from metabolic profiling and growth analysis. Comparison of intracellular amino acid concentrations in 2216E and in 1/10 2216E media. Two-tailed Student’s *t*-test was used to examine significant differences (* *p*-value < 0.05; ** *p*-value < 0.01; *** *p*-value < 0.001). The error bar represents the standard deviation of three biological replicates.

**Figure 7 ijms-24-08617-f007:**
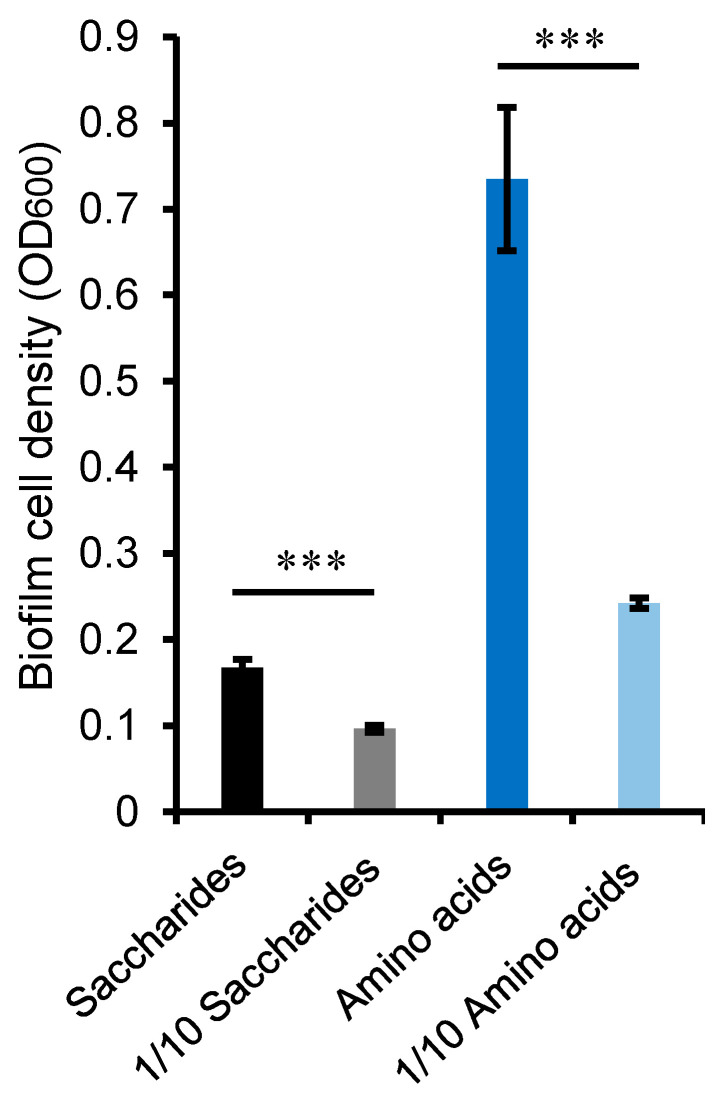
Maximum cell densities of M382 biofilms grown in saccharides, 1/10 saccharides, amino acids, and 1/10 amino acids. Two-tailed Student’s *t*-test was used to examine significant differences (*** *p*-value < 0.001). The error bar represents the standard deviation of three biological replicates.

**Figure 8 ijms-24-08617-f008:**
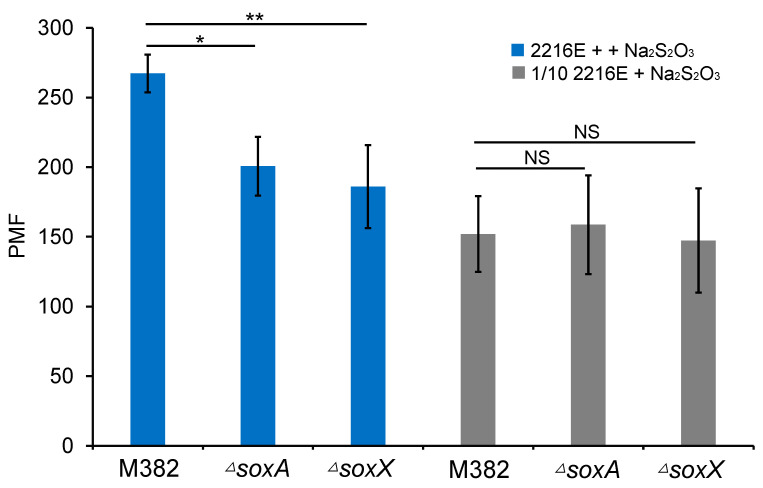
Comparison of proton motive force (PMF) between the biofilms of M382 and its two *sox* gene mutants grown in two concentrations of amino acids with thiosulfate. PMF was measured by the fluorescence intensity of DiSC3(5) after staining the cells. A two-tailed Student’s *t*-test was used to examine the significant differences (* *p*-value < 0.05; ** *p*-value < 0.01; NS, no significance). The error bars represent the standard deviation of the three biological replicates.

## Data Availability

The complete genome sequence of M382 and the transcriptomic datasets have been deposited in the NCBI database (BioProject number PRJNA753157).

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
