# Peer review of "Copiotrophy in a Marine-Biofilm-Derived Roseobacteraceae Bacterium Can Be Supported by Amino Acid Metabolism and Thiosulfate Oxidation"

_ijms, 2023, doi:10.3390/ijms24108617_

Round 1

Reviewer 1 Report

The work of Su and Zhang is interesting and brings a good contribution to the current status of knowledge. The work is of good quality I have no negative remarks except the fact that only minor editing errors require correction and that the conclusion section is missing.

Author Response

The work of Su and Zhang is interesting and brings a good contribution to the current status of knowledge. The work is of good quality I have no negative remarks except the fact that only minor editing errors require correction and that the conclusion section is missing.

Response: We have added this part in the revised manuscript. The authors appreciate your positive comments.

Conclusion: Together, the present study has clarified the mechanisms governing copiotrophy in a marine Roseobacteraceae bacterium. We showed that copiotrophy in this bacterium is the result of amino acid metabolism and the regulation of functionally relevant genes. These findings contribute to the understanding of marine bacterial lifestyles, as well as provide an explanation of bacterial adaptation in an ecological framework. Future perspectives would include a further study to illuminate the gene regulatory pathways or networks that control the response to the amino acid concentration, and a broad study to overview similar mechanisms among marine bacteria.

Thanks again for your comments.

Reviewer 2 Report

The manuscript by Su and Zhang examines the copiotrophic properties of marine isolate belonging to the order Rhodobacterales. The authors have applied various methods to analyze the growth strategies of this bacterium as a function of media concentration, including genomics, transcriptomics, and metabolomics, suggesting that copiotrophy in this bacterium may be supported by amino acid metabolism and thiosulfate oxidation. This work is comprehensive, but several major points must be addressed before publication in the journal IJMS.

-The authors should comment on using the nutrient source in terms of the effects obtained on amino acid metabolism. Is this effect the result of the carbon source used for cultivation? It is already known that amino acid metabolism is upregulated in complex media in E.coli (doi.org/10.1016/j.procbio.2017.04.003). Do you think the metabolism of bacteria would change if  grown on sugar instead of peptone and yeast extract as a carbon source?

-The use of taxonomy in this manuscript is somewhat confusing. According to the NCBI taxonomy, Roseobacter is a genus belonging to the family Rhodobacterales, order Roseobacteraceae. Ruegeria, on the other hand, is a different genus in the same family. Ruegeria is therefore not Roseobacter. This should be corrected throughout the manuscript, and Roseobacter should be replaced by Roseobacteraceae. Also, there should be a short explanation, of why the new species is classified as Rhodobacteraceae sp., belonging to the family Rhodobacteraceae, which is a different family than Roseobacteraceae, which the referent strains belong to.

-Figure 1. A bootstrap value of 50% is generally considered weak support for the branch and is not considered significant. This threshold should be at least 70%.

-Figures 2A and B. It seems that these plots are generated from only one experiment. It is necessary to repeat the experiment in order to verify these effects and provide new graphs that include standard deviations.

-Line 113. The difference obtained with the DSS-3 strain with two concentrations of media is not profound, please change this statement.

-Line 163. Are these bacteria closely related since they have only 74.46 % of identity?

-Sox gene knockout - some reference should be made to the method using conjugation between E.coli and the Roseobacteraceae strain for a gene knockout procedure, or the method should be explained in more detail, e.g.: the size of the flanking regions serving for homologous recombination, the number of positive clones expected and obtained after recombination, why sucrose is added as a marker for two-step recombination…so that the experiment can be repeated by others. 

Author Response

  1. The authors should comment on using the nutrient source in terms of the effects obtained on amino acid metabolism. Is this effect the result of the carbon source used for cultivation? It is already known that amino acid metabolism is upregulated in complex media in E. coli (doi.org/10.1016/j.procbio.2017.04.003). Do you think the metabolism of bacteria would change if grown on sugar instead of peptone and yeast extract as a carbon source?

Response: Thanks a lot for your constructive comments. To answer this question, we measured the growth of M382 in amino acids and saccharides. New results and discussion have been added.

Considering that bacterial amino acid metabolism can be up-regulated in complex media [Kim, J., & Kim, K. H. (2017). Effects of minimal media vs. complex media on the metabolite profiles of Escherichia coli and Saccharomyces cerevisiae. Process Biochemistry, 57, 64-71], we investigated M382 growth in the presence of amino acids and saccharides, to test the specificity of amino acid consumption. M382 biofilms were cultured in media with two concentrations of mixed amino acids (L-glutamic acid, glycine, L-tryptophan, L-alanine, L-threonine, L-cysteine, and L-serine) and the maximum cell densities were measured. In parallel, M382 biofilms were cultured with two concentrations of saccharides, which were used for comparison. As expected, after 24 hours of culture, the M382 biofilm grew to a cell density of OD600 = 0.74 in the 5 mM amino acid mixture and a cell density of OD600 = 0.24 in the 0.5 mM amino acid mixture (Fig. 7). The cell density of the M382 biofilm grown in 5 mM saccharides was OD600 = 0.167, compared to OD600 = 0.096 in the 0.5 mM saccharide-containing medium. Although significant differences in cell densities were seen with both amino acids and saccharides, the fold-change along with concentration variation was much greater for bacteria grown in amino acids (Fig. 7).

These results suggest a stronger influence of amino acid concentration on bacterial growth than the saccharide concentration was observed, which is also consistent with the notion that copiotrophic growth of M382 can be supported by amino acid metabolism.

  1. The use of taxonomy in this manuscript is somewhat confusing. According to the NCBI taxonomy, Roseobacter is a genus belonging to the family Rhodobacterales, order Roseobacteraceae. Ruegeria, on the other hand, is a different genus in the same family. Ruegeria is therefore not Roseobacter. This should be corrected throughout the manuscript, and Roseobacter should be replaced by Roseobacteraceae. Also, there should be a short explanation, of why the new species is classified as Rhodobacteraceae sp., belonging to the family Rhodobacteraceae, which is a different family than Roseobacteraceae, which the referent strains belong to.

Response: We agree with you. This bacterium has been renamed as “Roseobacteraceae sp. M382”, and it has been revised throughout the manuscript. Ruegeria pomeroyi DSS-3 also belongs to the family Roseobacteraceae.

  1. Figure 1. A bootstrap value of 50% is generally considered weak support for the branch and is not considered significant. This threshold should be at least 70%.

Response: As shown in Figure 1, the threshold has been reset to 70%.

  1. Figures 2A and B. It seems that these plots are generated from only one experiment. It is necessary to repeat the experiment in order to verify these effects and provide new graphs that include standard deviations.

Response: We did three biological replicates. We have added errors to indicate standard deviations in the revised Figure 2.

  1. Line 113. The difference obtained with the DSS-3 strain with two concentrations of media is not profound, please change this statement.

Response: Revised according to the comment. “After culture at 25 ℃ for 24 hours, M382 growth reached the stationary stage with OD600 values of approximately 1.4 and 0.4 for the original (labeled as 2216E) and 1/10 carbon concentrations (1/10 2216E), respectively, while the growth of DSS-3 was not profoundly affected by the carbon concentration (Fig. 2A,B).”

  1. Line 163. Are these bacteria closely related since they have only 74.46 % of identity?

Response: Although whole-genome average nucleotide identity (ANI) analysis revealed that M382 and R. pomeroyi DSS-3 shared only 74.46% identity, R. pomeroyi DSS-3, Ruegeria sp. THAF33, Ruegeria sp. AD91A, and Leisingera aquaemixtae R2C4, and Ruegeria sp. AD91A is closest relative of M382 that have been discovered so far.

  1. Sox gene knockout - some reference should be made to the method using conjugation between E.coli and the Roseobacteraceae strain for a gene knockout procedure, or the method should be explained in more detail, e.g.: the size of the flanking regions serving for homologous recombination, the number of positive clones expected and obtained after recombination, why sucrose is added as a marker for two-step recombination…so that the experiment can be repeated by others.

Response: The reference, as well as more details, have been added. Gene knockout was performed based on in-frame mutant construction and complementation [Fu, H.; Liu, L.; Dong, Z.; Guo, S.; Gao, H. Dissociation between Iron and Heme Biosyntheses Is Largely Accountable for Respiration Defects of Shewanella oneidensis fur Mutants. Appl Environ Microbiol. 2018, 84. http://doi.org/10.1128/AEM.00039-18], with modification. The size of the flanking regions serving for homologous recombination is 29 bp and these regions are marked red in Table S5. Finally, candidate bacterial cells were transferred to marine broth 2216E medium with 20% sucrose, where colonies with two-step recombination were generated and screened by PCR using the Up-F and Down-R primers. This step is to screen colonies that have successfully resected the suicide plasmid, which contains the sacB gene and cannot grow in sucrose. The number of successful clones obtained were approximately half the number of clones obtained in the first recombination.

Thanks again for your comments.

Reviewer 3 Report

Please see attached 

Author Response

  1. Line 124 - Figure 2 A and B - error bars are not visible. Were these also three biological replicates? Please specify.

Response: The authors appreciate your positive comments. We did three biological replicates. We have added errors to indicate standard deviations in the revised Figure 2.

  1. Line 146-147 - in terms of nitrogen metabolism - were the pathway for the conversion of ammonium to glutamine and glutamate also annotated? Are there glutamine synthetase genes found, which are considered to the central in the nitrogen metabolism of bacteria? Please clarify.

Response: Good question. We checked the KEGG annotation result, and found that M382 possessed the several copies of glutamine synthetase (glnA, K01915). We have added this result and revised Figure 3 accordingly.

  1. Line 186 - what compound exactly was the main carbon source in the tested medium - was it glucose at different concentrations?

Response: The carbon sources in 2216E medium is yeast extract and peptone. In the revised manuscript, we performed growth experiment using amino acids or saccharides as the carbon source. Please see Figure 7.

  1. Line 424 - please specify the length of homology arms used for recombination.

Response: The length of homology arms used for recombination is 29 bp. This regions are marked red in Table S5. Thanks again for your comments.

Round 2

Reviewer 2 Report

The authors have taken into account all the comments and the revised version of the paper has been significantly improved. I have one more minor comment, what are the "saccharides" that were used in the revised version of the paper? This should be indicated in M&M section.

Author Response

The authors have taken into account all the comments and the revised version of the paper has been significantly improved. I have one more minor comment, what are the "saccharides" that were used in the revised version of the paper? This should be indicated in M&M section.

Response: For the experiment using synthetic media with saccharides, equal concentrations of sucrose, L-glucose, D-glucose, L-fructose, D-fructose, D-maltose, and D-galactose, each at a concentration of 5 mM or 0.5 mM, were used as carbon sources. Again, inorganic nutrients were adjusted to the same concentrations. We have indicated this in the method part. Thanks.